

# Association of maternal genetics with the gut microbiome and eucalypt diet selection in captive koalas

Kotaro Kondo[1], Mirei Suzuki[1], Mana Amadaira[2], Chiharu Araki[2], Rie Watanabe[2], Koichi Murakami[3], Shinsaku Ochiai[3], Tadatoshi Ogura[2] and Takashi Hayakawa[4]

[1] Graduate School of Environmental Science, Hokkaido University, Sapporo, Hokkaido, Japan
[2] School of Veterinary Medicine, Kitasato University, Towada, Aomori, Japan
[3] Hirakawa Zoological Park, Kagoshima, Kagoshima, Japan
[4] Faculty of Environmental Earth Science, Hokkaido University, Sapporo, Hokkaido, Japan

Corresponding authors
Tadatoshi Ogura,
togura@vmas.kitasato-u.ac.jp
Takashi Hayakawa,
hayatak@ees.hokudai.ac.jp,
thawks.taxi@gmail.com

## ABSTRACT

**Background.** Koalas, an Australian arboreal marsupial, depend on eucalypt tree leaves for their diet. They selectively consume only a few of the hundreds of available eucalypt species. Since the koala gut microbiome is essential for the digestion and detoxification of eucalypts, their individual differences in the gut microbiome may lead to variations in their eucalypt selection and eucalypt metabolic capacity. However, research focusing on the relationship between the gut microbiome and differences in food preferences is very limited. We aimed to determine whether individual and regional differences exist in the gut microbiome of koalas as well as the mechanism by which these differences influence eucalypt selection.

**Methods.** Foraging data were collected from six koalas and a total of 62 feces were collected from 15 koalas of two zoos in Japan. The mitochondrial phylogenetic analysis was conducted to estimate the mitochondrial maternal origin of each koala. In addition, the 16S-based gut microbiome of 15 koalas was analyzed to determine the composition and diversity of each koala's gut microbiome. We used these data to investigate the relationship among mitochondrial maternal origin, gut microbiome and eucalypt diet selection.

**Results and Discussion.** This research revealed that diversity and composition of the gut microbiome and that eucalypt diet selection of koalas differs among regions. We also revealed that the gut microbiome alpha diversity was correlated with foraging diversity in koalas. These individual and regional differences would result from vertical (maternal) transmission of the gut microbiome and represent an intraspecific variation in koala foraging strategies. Further, we demonstrated that certain gut bacteria were strongly correlated with both mitochondrial maternal origin and eucalypt foraging patterns. Bacteria found to be associated with mitochondrial maternal origin included bacteria involved in fiber digestion and degradation of secondary metabolites, such as the families Rikenellaceae and Synergistaceae. These bacteria may cause differences in metabolic capacity between individual and regional koalas and influence their eucalypt selection.

**Conclusion.** We showed that the characteristics (composition and diversity) of the gut microbiome and eucalypt diet selection of koalas differ by individuals and regional origins as we expected. In addition, some gut bacteria that could influence eucalypt
foraging of koalas showed the relationships with both mitochondrial maternal origin and eucalypt foraging pattern. These differences in the gut microbiome between regional origins may make a difference in eucalypt selection. Given the importance of the gut microbiome to koalas foraging on eucalypts and their strong symbiotic relationship, future studies should focus on the symbiotic relationship and coevolution between koalas and the gut microbiome to understand individual and regional differences in eucalypt diet selection by koalas.

# INTRODUCTION

The marsupial koalas (*Phascolarctos cinereus*) live in eucalypt forests in eastern and southeastern Australia. They consume eucalypt tree leaves (*Eucalyptus* spp.) almost exclusively, which are potentially toxic and not suitable for other animals (*Moore & Foley, 2000*; *Shipley, Forbey & Moore, 2009*). They select and depend on a few eucalypt species and even conspecific eucalypt individuals. Although 120 eucalypt species have been recorded as koala food sources, each koala may only consume 1–10 eucalypt species. The Cape Otway population in Victoria reportedly experienced starvation and collapse of the individual number due to the high density of koalas (*Whisson et al., 2016*). The overbrowsing of their preferred trees (*E. viminalis*) resulted in the defoliation of these trees. Notably, another eucalypt tree species, *E. obliqua*, is preferred by some living koalas. Other koalas that preferred *E. viminalis* did not feed on *E. obliqua* and therefore suffered starvation, leading to their death. Thus, with regard to their extreme diet preferences, we aimed to determine the factors that lead to differences in food preferences among individual koalas.

Koalas have evident regional differences in morphology, such as size and color (*Melzer et al., 2000*; *Briscoe et al., 2015*), population density (*Whisson & Ashman, 2020*), and genetic diversity (*Johnson et al., 2018*). Due to these regional differences, the koalas are divided into northern (Queensland and northern New South Wales) and southern (Victorian and southern New South Wales) koalas for husbandry management and maintained under specific conditions (rearing temperature, feeding eucalypt species) suitable for each group (*Japanese Association of Zoos and Aquariums, 2020*). We here call these groups management groups. Previously, koalas were thought to have three subspecies based on morphological differences (*P. c. adustus*, *P. c. cinereus*, and *P. c. victor*) (*Melzer et al., 2000*; *Sherwin et al., 2000*). Currently, it is known that koalas can be divided into four distinct groups (two northern lineages, a central lineage, and a southern lineage) based on mtDNA analysis (*Neaves et al., 2016*). The genome-wide analysis also revealed that koalas are divided into five distinct groups (*Johnson et al., 2018*). Eucalypt vegetation potentially preferred by koalas varies from region to region (*Moore & Foley, 2000*). The nitrogen and fiber content and composition of potentially toxic plant secondary metabolites in eucalypts are known to vary by region and species (*Moore et al., 2004a*; *Moore et al., 2004b*; *Brice et al., 2019*).

Therefore, koalas have had to adapt to local vegetation in terms of chemistry and change their dietary preferences (*De Gabriel et al., 2010*).

The role and impact of symbiotic microorganisms on host animals have been highly recognized, including koalas (*Alberdi et al., 2016*; *Moeller & Sanders, 2020*). The gut microbiome contributes to many biological functions such as host metabolism (*Holmes et al., 2012*), detoxification of secondary metabolites (*Kohl et al., 2014*; *Zhu et al., 2018*; *Dearing & Weinstein, 2022*), immune system (*Hooper, Littman & Macpherson, 2012*; *Rooks & Garrett, 2016*) and behavior (*Leitão Gonçalves et al., 2017*; *Jia et al., 2021a*). The gut microbiome is particularly important in koalas, which use the enlarged hindgut to ferment eucalypts: highly fibrous, low in nutrition, and rich in secondary metabolites (*Cork, Hume & Dawson, 1983*; *Cork & Foley, 1997*). They have specifically developed their characteristics: morphology, such as large cecum and colon fermentation tanks (*Cork & Sanson, 1990*); physiology, such as adjustment of the speed of the passage of substances through the gastrointestinal tract according to their size (*Cork & Foley, 1997*); genetics, such as the expanded repertoire of bitter taste and olfactory receptor gene family and cytochrome P450 monooxygenase gene family (*Johnson et al., 2018*) to adapt to appropriate eucalypt selection, ability to digest and detoxify the leaves of eucalypts, leading to dependence on the gut microbiome (*Barker et al., 2013*; *Baker et al., 2017*). Koala juveniles consume their mother's feces, called pap, which contains a high concentration of microorganisms and digested eucalypt residues (*Osawa, Blanshard & Ocallaghan, 1993*); it enables juveniles to gain gut microorganisms necessary for their growth, development, and eucalypt digestion (*Minchin, 1937*). Despite such a strong relationship between the gut microbiome and eucalypt foraging in koalas, research focusing on the relationship between the gut microbiome and differences in food preferences is limited to only two studies in Cape Otway (*Brice et al., 2019*; *Blyton et al., 2019*).

Herein, we hypothesized that the adaptation of the gut microbiomes of koalas to region-specific eucalypt vegetation is associated with regional differences in eucalypt selection. A previous study reported the food preferences of individual koalas from Japanese zoos (*Ogura et al., 2019*). Thus, we aimed to reveal whether there is difference in gut microbiome and eucalypt diet selection among koalas from different regional origins. We investigated the maternal genetics (mitochondrial lineage and gut microbiome) of 15 captive koalas in Japan. Subsequently, we investigated the relationship among origin (mitochondrial lineage and management group), eucalypt diet selection, and 16S-based gut microbiomes in captive koalas from two Japanese zoos.

## MATERIALS & METHODS

### Ethics statement

This study adhered to the Animals in Research: Reporting On Wildlife (ARROW) guidelines. Sample collection and behavioral recordings were approved by Hirakawa Zoological Park and Awaji Farm Park England Hill as collaborative projects with Hayakawa Lab of Hokkaido University and Ogura Lab of Kitasato University and were fully performed through noninvasive approaches, except for blood collection. To minimize suffering,

blood samples were not collected for the purpose of this study; instead, used residues from routine health examinations were employed. Behavioral recordings were completely noninvasively conducted in the zoo-visitors' area and did not artificially control koalas' behavior for the purpose of this study. The koalas were healthily kept for the purpose of public exhibitions in the zoos in the enough space of enclosure (>5 m wide, >5 m depth, and >5 m height). Their environments were enriched in terms of their active movement as follows. They were always provided branches of a variety of eucalypt species scattered in the enclosure to increase their active movement to find the eucalypt diet items. Logs were arranged in a three-dimensional manner for them to enjoy moving in any direction. In Japan, different management groups are fed different types of eucalypts according to their preferences, in accordance with husbandry guidelines issued by Japanese Association of Zoos and Aquariums (*Japanese Association of Zoos and Aquariums, 2020*). The animal experimentation protocol was approved by the President of Kitasato University through the judgment of the Institutional Animal Care and Use Committee of Kitasato University (Approval No. 21-069).

## Animals

To perform sufficient statistical analysis, a total of 15 captive koalas were selected and examined in this study (Table 1, Table S1). Of these, nine koalas obtained care at Hirakawa Zoological Park, whereas the other six obtained care at Awaji Farm Park England Hill. Four of the Awaji koalas were southern koalas in the management grouping (Yuki, Daichi, Nozomi, and Midori), whereas the others (Peta and Umi) and all koalas in Hirakawa Zoological Park were northern koalas in the management grouping. We selected these two zoos because Hirakawa Zoological Park has large number of koalas and Awaji Farm Park England Hill is the only zoo that keeps southern koalas in the management grouping in Japan. To be blind test, behavioral recording was performed by MA, CA and RW, and statistical analysis was performed by KK.

## Mitochondrial phylogenetic analysis

Mitochondrial phylogenetic analysis, based on the study by *Neaves et al. (2016)*, was conducted to estimate the mitochondrial maternal origin of each koala. Blood or fecal samples of koalas were collected to extract and purify genomic DNA. The collected blood samples were immediately mixed with anticoagulants (EDTA or heparin). All samples were frozen at −20 °C prior to DNA extraction. Total DNA was extracted from blood samples using Qiagen DNeasy Blood and Tissue Kit (Qiagen GmbH, Hilden, Germany) and from fecal samples using QIAamp Fast DNA Stool Mini Kit (Qiagen GmbH). Next, using TaKaRa Ex Taq Hot Start Version (Takara Bio Inc., Shiga, Japan), the mitochondrial DNA control region (D-loop) was amplified *via* polymerase chain reaction (PCR) using the following primers: MaL15999M (ACC ATC AAC ACC CAA AGC TGA) and MaH16498M (CCT GAA GTA GCA ACC AGT AG) (Fumagalli et al. 1997). The PCR conditions were as follows: initial denaturation (94 °C for 10 min); followed by 35 cycles of denaturation (94 °C for 10 s), annealing (60 °C for 30 s), and extension (72 °C for 60 s); and final extension (72 °C for 10 min). The PCR products were purified *via* precipitation with isopropanol.

**Table 1  Data on koalas included in this study.**

| Zoo | Name | Sex | Year | Foraging | mtDNA | Management | Number of fecal samples |
|---|---|---|---|---|---|---|---|
| Hirakawa | Boonda | M | 11 | + | Northern 2 | Northern | 3 |
| | Himawari | F | 2 | + | Southern | Northern | 2 |
| | Itsuki | M | 1 | + | Central | Northern | 3 |
| | Ito | F | 4 | + | Central | Northern | 4 |
| | Kibou | F | 2 | + | Central | Northern | 4 |
| | Sora | M | 1 | + | Southern | Northern | 3 |
| | Indeco | F | 2 | − | Southern | Northern | 5 |
| | Archer | M | 3 | − | Southern | Northern | 5 |
| | Peace | F | 1 | − | Central | Northern | 3 |
| Awaji | Yuki | M | 13 | − | Southern | Southern | 5 |
| | Daichi | M | 8 | − | Southern | Southern | 5 |
| | Nozomi | F | 14 | − | Southern | Southern | 5 |
| | Midori | F | 25 | − | Southern | Southern | 5 |
| | Umi | F | 8 | − | Northern 2 | Northern | 5 |
| | Peter | M | 6 | − | Southern | Northern | 5 |

**Notes.**
The scores in Foraging ("+" or "−") indicates individuals, where the foraging data have been collected ("+") or no foraging data have been collected ("−"). The mtDNA shows the mitochondrial lineage of each individual. The Management shows the management group of each koala.

Next, the purified PCR products were directly sequenced using PCR primers for complete coverage in both strand orientations *via* BigDye Terminator v3.1 Cycle Sequencing Kit and 3130 Genetic Analyzer (Applied Biosystems, Bedford, MA, USA). Chromatograms were imported into FinchTV (Geospiza Inc., Seattle, WA, USA) and analyzed.

The phylogeny of the sequenced D-loop region of mitochondrial DNA was analyzed with the sequences of 48 koalas reported by *Neaves et al. (2016)*. A multiple alignment was constructed by MUSCLE (*Edgar, 2004*). A phylogenetic tree was reconstructed using the maximum-likelihood (ML) method with 1,000 bootstrap resamplings using MEGA11 (*Tamura, Stecher & Kumar, 2021*).

## Foraging data and gut microbiome

Foraging data were collected from six koalas in Hirakawa Zoological Park for 8 days between 21–30 November 2021. Subjects were housed independently (Boonda and Ito) or in cohabitation with two individuals (Himawari with Kibou and Sora with Itsuki). During foraging and feeding, all koala pairs that use the same enclosure rarely interfere with each other. At Hirakawa, four of the five eucalypt species (*E. camaldulensis*, CR; *E. microcorys*, M; *E. punctata*, P; *E. robusta*, R; *E. tereticornis*, T) were fed twice a day (9:00 and 16:00). Since Japan is not a natural habitat of eucalypts, available eucalypts for koala diet are limited depending on the cultivation and logistics (*Japanese Association of Zoos and Aquariums, 2020*). Eucalypt species fed to koalas in this study were regularly fed to koalas as their typical food as available in these zoos. The combination of eucalypts fed at the same time and the frequency of feeding was counterbalanced. The method of behavioral recoding

was determined *a priori*. Eucalypt species consumed by each koala were observed using the instantaneous focal sampling method in 30-s intervals (*Altmann, 1974*). The observation time was 1 h each at 9:00 am and 4:00 pm immediately after feeding by caretakers and 1 h each at 11:00 am and 2:00 pm outside of the immediate feeding period. Finally, the observation was carried out for 8 days (a total of 32 h per individual).

Fecal samples were collected during foraging observation periods. Fecal sampling was performed by collecting fresh feces immediately after defecation. In case of cohabiting housed individuals, individual identification was performed by direct observation of defecation. The fecal samples of nonsubject koalas were also collected in Hirakawa and Awaji for comparison. Fresh fecal samples were collected for sampling. Finally, 2–5 fecal samples per koala and a total of 62 fecal samples were collected and stored at −20 °C until DNA extraction.

### 16S rRNA gene sequencing

According to *Hayakawa et al. (2018a)*; *Hayakawa et al. (2018b)*, we performed 16S-based gut microbiome analysis using collected fecal samples. After quantifying the concentration of purified fecal DNA using Qubit dsDNA HS Assay Kit equipped with Qubit fluorometer (Thermo Fisher Scientific, Waltham, MA, USA), we amplified the V3–V4 region of the 16S rRNA gene using KAPA HiFi Hot Start Ready Mix (Kapa Biosystem, Inc., Wilmington, DE, USA). We used the following primer pair: 1S-D-Bact-0341-b-S-17 (forward), CCT ACG GGN GGC WGC AG; S-D-Bact-0785-a-A-21 (reverse), GAC TAC HVG GGT ATC TAA TCC (*Klindworth et al., 2013*), with the specific overhang adaptors TCG TCG GCA GCG TCA GAT GTG TAT AAG AGA CAG-[3-6-mer Ns]-[forward primer] and GTC TCG TGG GCT CGG AGA TGT GTA TAA GAG ACA G-[3-6mer Ns]-[reverse primer], where 3-6mer Ns can improve the sequencing quality (*Lundberg et al., 2013*). After confirming PCR amplification *via* gel electrophoresis, we purified the PCR amplicons with Agencourt AMPure XP beads (Beckman Coulter, Inc, Brea, CA, USA). Then, we performed index PCR using Illumina Nextera XT Index Kit (Illumina, Inc., San Diego, CA, USA). Additionally, we confirmed the presence and appropriate length of index PCR products *via* electrophoresis and then purified index PCR products with Agencourt AMPure XP beads. All index PCR products were mixed at the same molarities for constructing a library. After PhiX spike-in (30%), we sequenced the library using the Illumina Miseq platform (Illumina, Inc., San Diego, CA, USA) (2 × 301 bp).

### Data analysis

The MiSeq base calls were converted to FASTQ files using configureBclToFastq.pl implemented by the bcl2fastq conversion software v1.8.4 (Illumina, Inc., San Diego, CA, USA) (options: no-eamss, mismatches 0, and use-bases-mask Y300n,Y8,Y8,Y300n). The read pairs were demultiplexed, the primer sequences were trimmed, and those with low-quality index sequences were discarded, where the index sequences included nucleotide(s) with a quality score of <30, using clsplitseq in Claident (*Tanabe & Toju, 2013*) (option: minqualtag = 30).

Quality control and data analysis were performed using QIIME2 v2021.4.0 (*Caporaso et al., 2010*). To generate amplicon sequence variants (ASVs), DADA2 v2021.4.0 (*Callahan et*

*al., 2016*) was used to quality filter the sequences with a read cut length of 260 (forward) and 260 (reverse) based on quality control results and denoise chimeric sequences with a read count of 1,000,000 for training the error model. The taxonomy analysis was conducted with the SILVA 138 database (*Bokulich et al., 2018*; *Robeson 2nd et al., 2021*). Subsequently, sample depths were rarefied where the value was $\geq$0.99 for all samples using Good's coverage (4,105 sequences per sample). Microbial composition of the samples is shown in Fig. S1 at phylum level and at genus level.

We determined alpha diversity (Shannon index, Chao1, Simpson index, Simpson index of evenness, Pielou's evenness index, Faith's phylogenetic diversity, and observed features) and beta diversity (unweighted UniFrac, weighted UniFrac (*Lozupone & Knight, 2005*; *Lozupone et al., 2007*), Jaccard index, and Bray–Curtis dissimilarity) to analyze differences in the gut microbiome between mitochondrial lineages or between the management groups. The pairwise Kruskal–Wallis test with Benjamini–Hochberg correction was used for comparing alpha diversity, whereas permutational multivariate analysis of variance (PERMANOVA) was used to assess the effect of mitochondrial lineage on gut microbiome similarity. Individual foraging data (proportion of foraging of each eucalypt species) were used to visualize similarities in foraging patterns using nonmetric multidimensional scaling (NMDS) *via* vegan v2.6.4 in R v4.2.3 (*R Core Team, 2023*). PERMANOVA was performed using the vegan package to assess the effect of mitochondrial lineage on foraging patterns.

To investigate the relationship between the diversity of individual eucalypt foraging and alpha diversity of the gut microbiome, Spearman's rank correlation coefficients between the alpha diversity (Shannon index) of the foraging and alpha diversity of the gut microbiome were calculated. In this analysis, the Shannon index was used to indicate the diversity of individual eucalypt foraging. Analysis of composition of microbiomes (ANCOM) (*Mandal et al., 2015*) was used to determine whether gut bacteria with characteristic relative abundances exist in each mitochondrial lineage or management group. All codes are available in the Supplemental Information.

## RESULTS

### Mitochondrial phylogeny

We performed phylogenetic analysis of the mitochondrial D-loop to estimate the mitochondrial maternal origins of the analyzed koalas (Fig. 1). A phylogenetic tree was constructed using the sequences of five koala haplotypes from Japanese zoos as well as the sequences of 48 individuals used for phylogenetic analysis by *Neaves et al. (2016)*. Thus, we revealed that koalas from Japanese zoos examined in this study have three different origins (northern-2, $N = 2$; central, $N = 4$; and southern, $N = 9$).

### Gut microbiome diversity among mitochondrial lineages

We investigated the mechanism by which the mitochondrial maternal origins of koalas, which were determined using mitochondrial phylogenetic analysis, and management groups influence the gut bacterial microbiome (Dataset S1, S2). Seven indices of alpha diversity were calculated (Tables S2, S3, Fig. 2). The index that considers evenness revealed that alpha diversity was the highest in the southern lineage and lowest in the central lineage

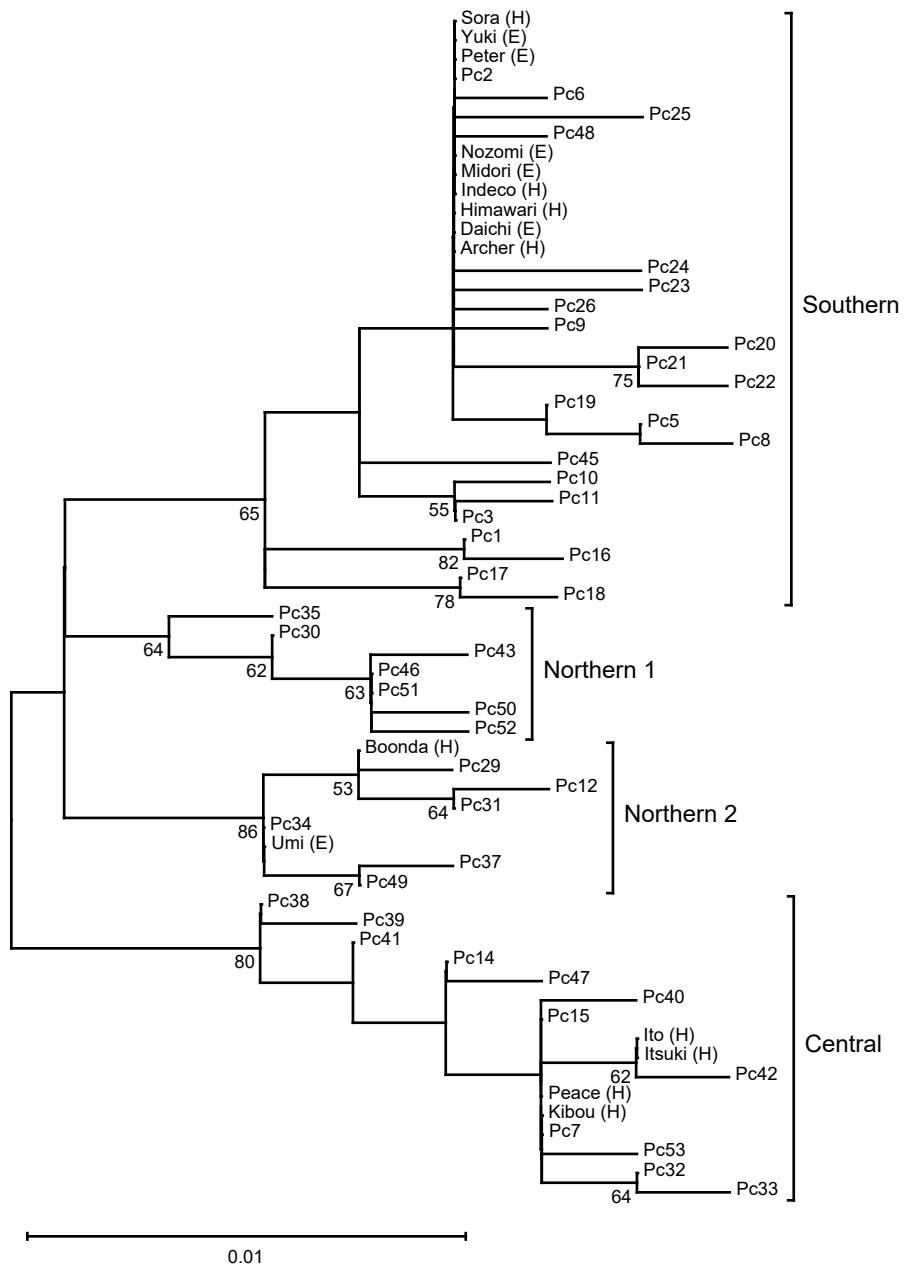

**Figure 1  The ML tree of the D-loop in koalas.** This is inferred using the Tamura three-parameter model with a discrete Gamma distribution (five categories), which has the lowest Bayesian Information Criterion scores (unrooted). The tree with the highest logarithmic likelihood (−990.84) is shown. The percentage (≥ 50%) of trees based on 1,000 bootstrap replications is shown below the branches. This tree shows four clades (Northern 1, Northen 2, Central, and Southern), which corresponding to the clades found by *Neaves et al. (2016)*.

(Simpson Index; South = 0.87 ± 0.07, Center = 0.75 ± 0.08, North2 = 0.84 ± 0.04; Simpson Index of Evenness; South = 0.10 ± 0.04, Center = 0.04 ± 0.01, North2 = 0.06 ± 0.01). In contrast, the index considering richness revealed that the southern lineage had

the lowest alpha diversity (Chao1; South = $103.55 \pm 28.36$, Center = $129.84 \pm 19.18$, North2 = $125.31 \pm 9.13$; Faith's phylogenetic diversity; South = $6.78 \pm 0.98$, Center = $7.81 \pm 0.92$, North2 = $7.50 \pm 0.54$; and observed features; South = $91.95 \pm 21.58$, Center = $115.57 \pm 18.18$, North2 = $113.25 \pm 5.36$). This trend was more clearly demonstrated between the management groups (Figs. 2A, 2B) than the mitochondrial lineages (the pairwise Kruskal–Wallis test with Benjamini–Hochberg correction; Shannon Index; South = $4.32 \pm 0.22$, North = $3.96 \pm 0.50$, $P = 0.002022$; Simpson Index; South = $0.90 \pm 0.02$, North = $0.81 \pm 0.09$, $P = 2.60E\text{-}07$; Simpson Index of Evenness; South = $0.14 \pm 0.03$, North = $0.06 \pm 0.02$, $P = 4.12E\text{-}10$; Chao1; South = $89.18 \pm 11.66$, North = $123.30 \pm 25.96$, $P = 0.000003$; Faith's phylogenetic diversity; South = $6.28 \pm 0.50$, North = $7.50 \pm 0.98$, $P = 0.000001$; and observed features; South = $79.80 \pm 6.94$, North = $109.67 \pm 20.61$, $P = 2.28E\text{-}07$).

Mitochondrial lineage have a significant impact on gut microbiome similarity (beta diversity), both qualitatively (Unweighted UniFrac) and quantitatively (weighted UniFrac) Management groups have a significant impact on gut microbiome similarity, both qualitatively and quantitatively (pairwise PERMANOVA tests; Unweighted UniFrac, pseudo-$F = 16.698158$, $q = 0.001$; number of permutations = 999, Fig. 3A; weighted UniFrac, pseudo-$F = 27.86$, $q = 0.0015$; number of permutations = 999, Fig. 3B). In the same way, mitochondrial lineages also have a significant impact (pairwise PERMANOVA tests; Unweighted UniFrac, Central *vs* Northern 2: pseudo-$F = 8.03$, $q = 0.001$, Central *vs* Southern: pseudo-$F = 8.13$, $q = 0.001$, Northern 2 *vs* Southern: pseudo-$F = 7.29$, $q = 0.001$; number of permutations = 999, Fig. 3C; weighted UniFrac, Central *vs* Northern 2: pseudo-$F = 5.78$, $q = 0.0015$, Central *vs* Southern: pseudo-$F = 11.22$, $q = 0.0015$, Northern 2 *vs* Southern: pseudo-$F = 5.09$, $q = 0.002$; number of permutations = 999, Fig. 3D).

Based on differences in gut microbiome similarity between mitochondrial lineages, we investigated whether there were bacteria that differed in relative abundance with mitochondrial lineages or management groups using ANCOM. The ANCOM results revealed significant differences in the relative abundance of 12 bacterial genera between the two management groups (Table S4, Fig. 3E). Of them, eight genera showed higher relative abundance in southern koalas, whereas the four other genera showed higher relative abundance in northern koalas. Further, two bacterial genera showed significant differences among mitochondrial lineages (Table S5, Fig. 3F). These two genera clearly showed differences among mitochondrial lineages; they are unknown genera belonging to the families *Tannerellaceae* (significant in southern, $W = 119$) and *Rikenellaceae* (significant in northern 2, $W = 111$).

## Relationship between eucalypt foraging and the gut microbiome

We recorded foraging data on zoo koalas to investigate the relationship between eucalypt foraging and the gut microbiome (Fig. 4A; raw data in Dataset S2). The proportion of each eucalypt species in foraging varied greatly from individual to individual. These foraging data were used to visualize the similarity of NMDS foraging patterns (Fig. 4B). PERMANOVA

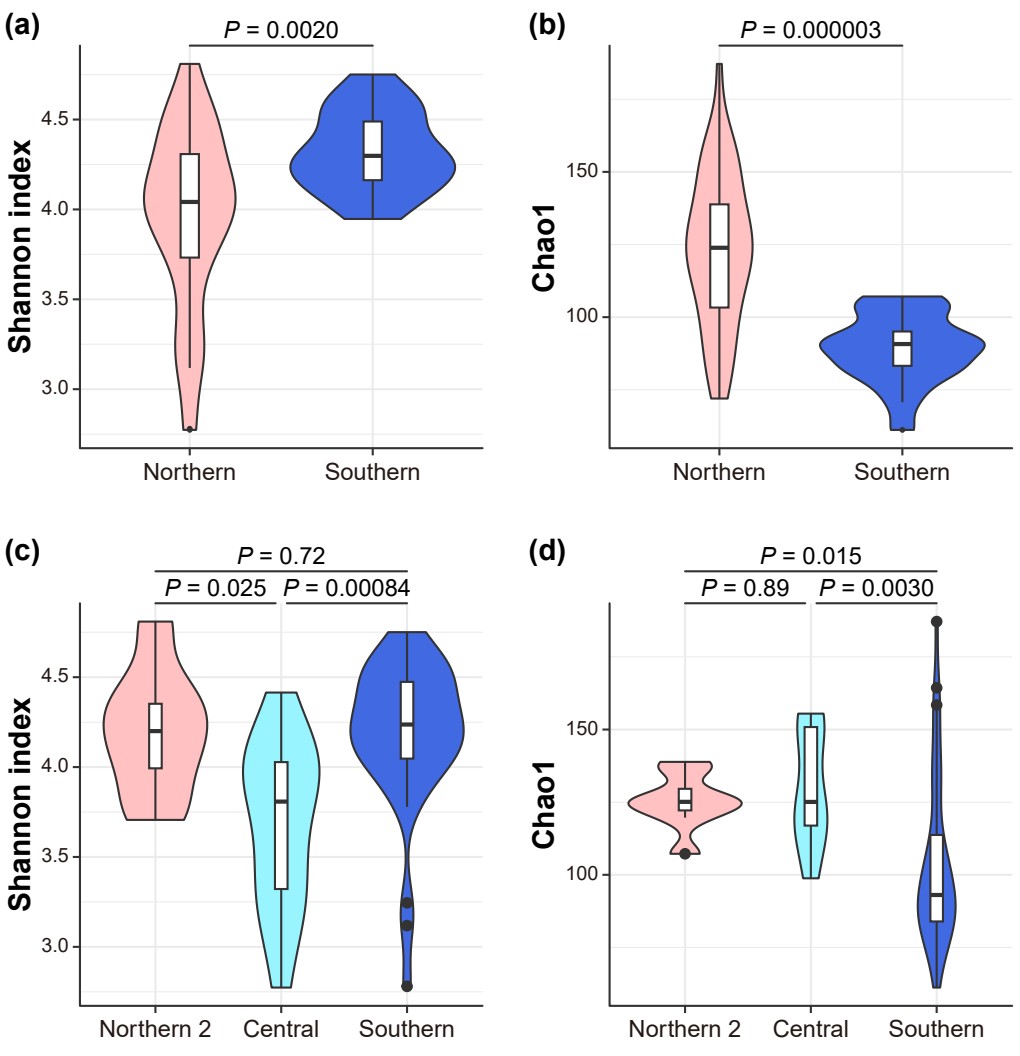

**Figure 2 Alpha diversity.** The differences between management groups in (A) Shannon index and in (B) Chao1 and mitochondrial lineages in (C) Shannon index and (D) Chao1. Statistical tests are conducted by the pairwise Kruskal–Wallis test with Benjamini–Hochberg correction.

showed that mitochondrial lineage significantly influences foraging patterns ($F = 5.88$, $R^2 = 0.68$, $P = 0.014$, number of permutations = 719).

We investigated the relationship between the diversity of eucalypt foraging and the gut microbiome diversity *via* Spearman's rank correlation coefficients (Table S6, Fig. 4C). The result shows a positive correlation between the diversity of the foraging and diversity of the gut microbiome (Spearman's rank correlation; $\rho = 0.89$, $P = 0.009$, Shannon entropy). There was a correlation with the indicator that considers evenness but not with the indicator that considers richness (Table S6), *i.e.*, significant in Shannon, Simpson, Simpson's index of evenness, and Pielou's evenness index, but not in Chao 1, Faith's phylogenetic diversity and observed features (Spearman's rank correlation; Shannon entropy: $\rho = 0.886$, $P = 0.009$, Pielou evenness: $\rho = 0.770$, $P = 0.036$, Simpson: $\rho = 0.830$,

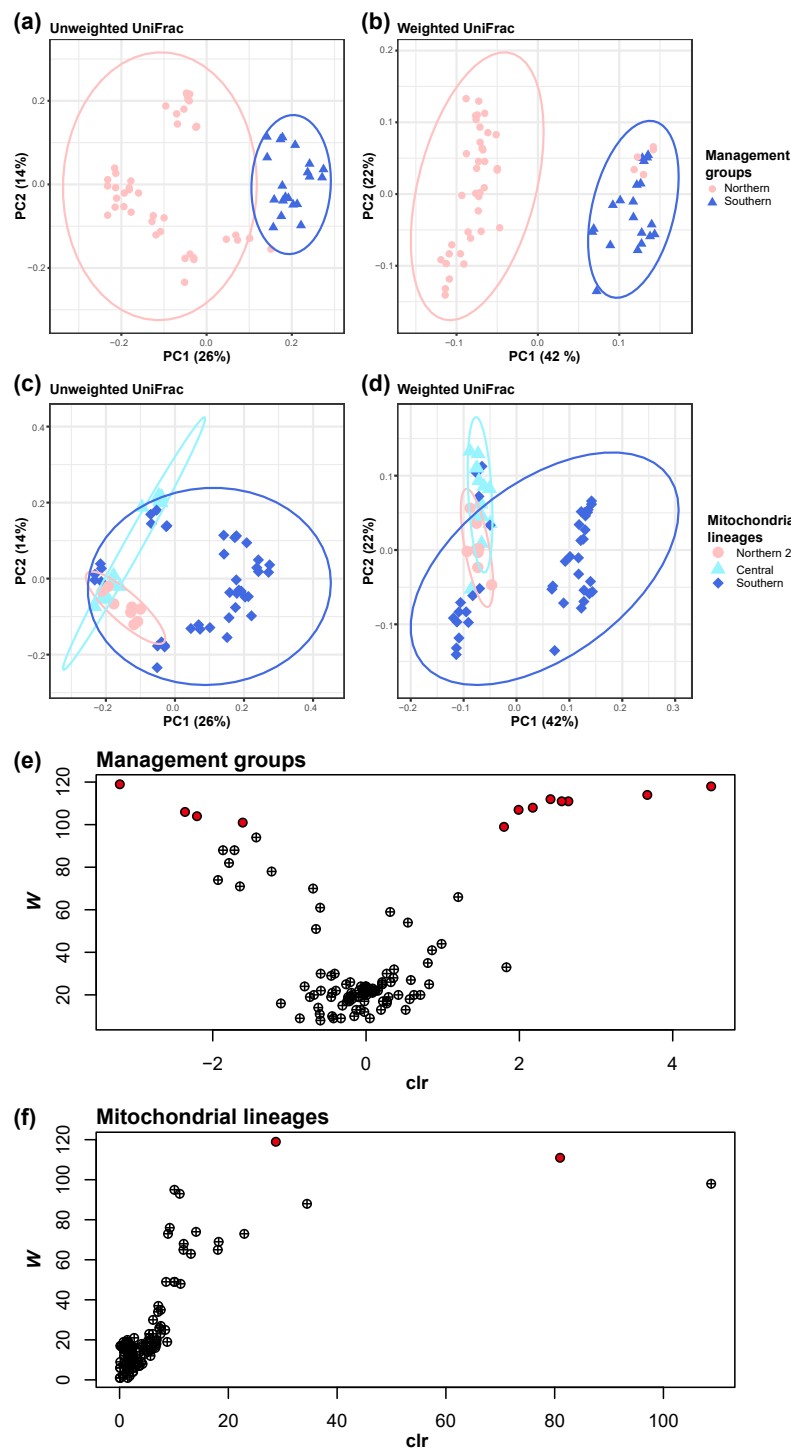

**Figure 3** **Management groups and mitochondrial lineages significantly influenced the gut bacterial community.** Clustered by the management groups in (A) unweighted UniFrac (pairwise PERMANOVA tests: pseudo- $F = 16.698158$; $P = 0.001$; number of permutations = 999) and (B) weighted UniFrac (pairwise PERMANOVA tests: pseudo- $F = 27.864924$; $P = 0.001$; 

**Figure 3 (…continued)**
number of permutations = 999). Clustered by mitochondrial lineages in (C) unweighted UniFrac (pairwise PERMANOVA tests, Center *vs* North2: pseudo- $F = 8.025514$; $q = 0.001$; Center *vs* South: pseudo- $F = 8.133812$; $q = 0.001$; North2 *vs* South: pseudo- $F = 7.289387$; $q = 0.001$; number of permutations = 999) and (D) weighted UniFrac (pairwise PERMANOVA tests, Center *vs* North2: pseudo- $F = 5.784462$; $q = 0.0015$; Center *vs* South: pseudo- $F = 11.216916$; $q = 0.0015$; North2 *vs* South: pseudo- $F = 5.085741$; $q = 0.002$; number of permutations = 999). (E) Volcano plot of the results of the analysis of the composition of microbiomes (ANCOM) between management groups or (F) mitochondrial lineages at the genus level. Each circle represents a taxon. Those with statistically significant differences based on the *W* statistics between mitochondrial lineages are colored red.

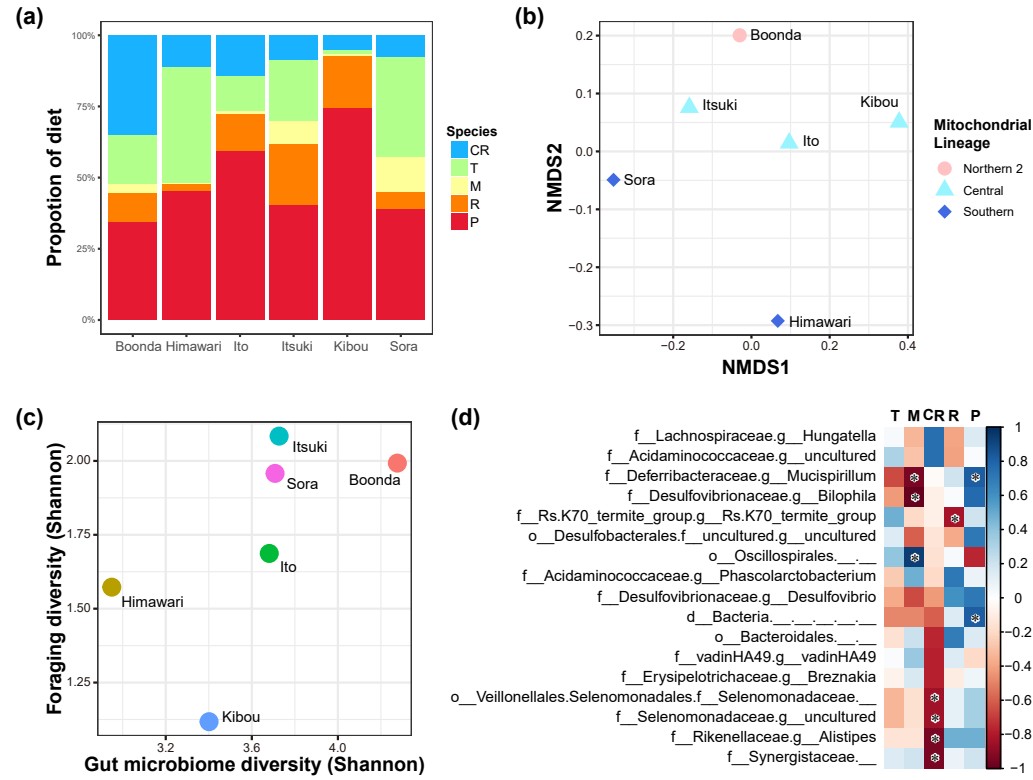

**Figure 4  Relationships of diet and gut microbiome.** (A) Diet proportion of each individual. (B) Results of NMDS using foraging data. Mitochondrial lineages significantly influenced the foraging pattern ($F = 5.88$; $R^2 = 0.677$; $P = 0.014$; number of permutations = 719). (C) Correlation between the foraging diversity and gut microbiome diversity (Shannon index). A significant correlation was observed between foraging diversity and gut microbiome diversity (Spearman's rank correlation; $\rho = 0.886$, $P = 0.009$). (D) The heat map shows changes in the relative abundance of gut bacteria because of changes in the proportion of eucalypt species in the diet (the color bar shows the strength of correlation). Spearman's rank correlation was conducted with Benjamini–Hochberg correction, and bacteria with significant correlations are indicated with an asterisk ($q < 0.05$; see Table S7 for individual values). The genera with the correlation coefficient $\|\rho\|$ of > 0.7 are shown. T, *E. tereticornis*; M, *E. microcorys*; CR, *E. camaldulensis*; R, *E. robusta*; P, *E. punctata*.

$P = 0.021$, Simpson e: $\rho = 0.830$, $P = 0.021$, Chao1: $\rho = 0.030$, $P = 0.479$, Faith pd: $\rho = 0.540$, $P = 0.133$, Observed features: $\rho = 0.030$, $P = 0.479$).

We investigated whether gut bacteria at the genus level vary in relative abundance depending on the foraging proportion of the eucalypt species. We revealed that the relative abundance of some gut bacteria increased or decreased depending on the foraging proportion of each eucalypt species (Table S7, Fig. 4D). The number of gut bacterial genera that showed a significant positive correlation was one with *E. microcorys* and two with *E. punctata* (Spearman's rank correlation with Benjamini–Hochberg correction; $q < 0.05$). Conversely, the number of gut bacterial genera that showed a significant negative correlation was four with *E. camaldulensis*, two with *E. microcorys,* and one with *E. robusta* (Spearman's rank correlation with Benjamini–Hochberg correction; $q < 0.05$).

## DISCUSSION

As the koala gut microbiome is known to be highly involved in eucalypt digestion and detoxification (*Baker et al., 2017*; *Osawa, Blanshard & Ocallaghan, 1993*), there is an increasing need to consider the gut microbiome in koala management, both *in situ* (within habitats) and *ex situ* (outside habitats) conservation (*Littleford-Colquhoun et al., 2022*). In general, the abundance and composition of the gut microbiome varies in response to dietary changes (*Scott et al., 2013*). However, the koala gut microbiome has been reported to be stable within individuals (*Eisenhofer et al., 2023*) and does not change significantly in response to dietary changes (*Blyton et al., 2019*; *Blyton et al., 2023*). Furthermore, there is a report that geographic distance of habitat has an influence on the similarity of the gut microbiome of wild koalas (*Littleford-Colquhoun et al., 2022*). Thus, it is considerable that this stable regional difference in the koala gut microbiome could be driving a significant difference in eucalypt selection between regions. However, research focusing on the relationship between the gut microbiome and differences in dietary preferences and its regionality is limited. Therefore, in this study we sought to elucidate how regional differences influence koalas' gut microbiomes and their eucalypt selection, and the mechanism by which these regional differences influence eucalypt selection. As a result, we showed presence of the relationship between the gut microbiome and eucalypt diet selection as well as the effect of geographical distribution on both factors by analyzing foraging data, mitochondrial lineages as indicators of regional origin, and the gut microbiome in captive koalas in Japan.

We demonstrated the influence of mitochondrial lineages on foraging patterns in eucalypt diet selection (Fig. 4B). Additionally, differences in the alpha and beta diversity values of the gut microbiome were observed between mitochondrial lineages and management groups (Table S2, Figs. 2A, 2B, 3A–3D). These results indicate regional variations in the koala gut microbiome and eucalypt diet selection, suggesting that different gut microbiomes in different regions may lead to variation in eucalypt diet selection. It is well known that joey koalas are fed with cecum feces, known as pap, of their mothers as an important weaning diet, after which they acquire the gut microbiome necessary for eucalypt foraging (*Osawa, Blanshard & Ocallaghan, 1993*; *Blyton et al., 2022*). The gut

microbiome was more similar between a mother and her offspring than between the father or other individual (Fig. S2). For example, the gut microbiome of Ito was very similar to her son, Itsuki, but not so similar to her father, Boonda. Additionally, mitochondrial DNA is inherited from the mother to child (*Giles et al., 1980*). Genetic factors such as detoxification and bitter taste receptor genes can also affect eucalypt diet selection (*Johnson et al., 2018*). Thus, the genetic inheritance in koalas through vertical transmission may explain the differences in eucalypt foraging and the gut microbiome.

Investigating the relationship between the diversity of gut microbiome and the diversity of koala eucalypt foraging, we found that the more diverse gut microbiome the koalas had, the more diverse eucalypts that they ate (Fig. 4C). Only the diversity of the gut microbiome, considering evenness rather than richness, was found to be correlated with the diversity of the foraging of eucalypts (Table S5). This finding suggests that the uniformity of the gut microbiome, rather than the presence of many bacteria, is more important for the foraging of diverse eucalypts. Since there were differences in evenness of alpha diversity among mitochondrial lineages and management groups (both highest in the southern lineage), these differences may explain the different food habits of koalas by region, *i.e.,* this may influence the number of eucalypt species preferred in different regions.

It is known that small animals need to selectively consume a high-quality diet while large animals need to consume large amounts because their energy requirements, tolerance to toxic substances, and amount of fermentable fiber vary with their body size and organ size (*Geist, 1974*; *Gaulin, 1979*; *Takatsuki & Padmalal, 2009*). According to Bergmann's rule, there are geographical variations in koala body size, with northern individuals known to be smaller and southern individuals larger as Australia is in the southern hemisphere and temperatures are cooler in the south (*Melzer et al., 2000*; *Briscoe et al., 2015*). Therefore, small northern individuals likely need to selectively consume eucalypt leaves, whereas larger southern individuals need to consume large amounts of eucalypt leaves. Thus, this need for optimal selection of foraging by body size may have led to the selection of a gut microbiome that allows northern individuals to consume specific types of eucalypts (a composition suitable for consuming specific eucalypts) and a gut microbiome that allows southern individuals to consume many eucalypts (highly diverse, especially even, gut microbiome) (Fig. 4C).

Previous studies have reported that gut bacteria influence the host diet (*Dearing & Weinstein, 2022*). For example, the gut bacteria reportedly contribute to oxalate degradation in the creosote bush diet of woodrats (*Neotoma spp.*) (*Kohl et al., 2014*) as well as mimosine degradation in the *Leucaena leucocephala* diet of Australian cattle (*Pratchett & Jones, 1991*; *Derakhshani, Corley & Al Jassim, 2016*), thus influencing diet. In koalas, differences in the composition of the gut microbiome may explain different feeding habits, revealing that fecal inoculation alters feeding habits (*Brice et al., 2019*; *Blyton et al., 2019*). The current study revealed that the relative abundance of several gut bacteria in koalas was associated with mitochondrial lineages, management groups, and foraging proportion of each eucalypt species (Fig. 4D). The family *Rikenellaceae* was associated with the northern lineage 2. This family is known to be involved in carbohydrate degradation (*Rowland et al., 2018*). In addition, the genus RC9 group of *Rikenellaceae* is known to play an important role in

crude fiber digestion (*Qiu et al., 2022*). The genus *Parabacteroides* was abundantly detected in southern koalas in the management group and has been reported to possess many oligosaccharide-degrading genes and genes associated with tannin degradation (*Moore et al., 2004b*). The family *Synergistaceae* was abundantly found in southern koalas in the management group. This family is known to be involved in the degradation of secondary plant metabolites (*Allison et al., 1992*). These bacteria may lead to differences in the metabolic capacities of individuals and affect the foraging patterns of different individuals and regions. Because 16S rRNA gene sequencing data is compositional data and thus difference in relative abundance of these taxa might be influenced by the compositionality, these results should be treated with caution (*Gloor et al., 2017*).

Although 16S rRNA gene sequencing performed in this study provides information on the composition and taxonomy of the gut microbiome (*Gołębiewski & Tretyn, 2020*), the relevant functional information is limited (*Dearing & Weinstein, 2022*). As duplicate functions (redundancy) of bacteria have been reported in other species (*Moya & Ferrer, 2016*; *Louca et al., 2018*), studies have also suggested a redundancy in the gut bacterial function of koalas (*Littleford-Colquhoun et al., 2022*; *Eisenhofer et al., 2023*). Therefore, we believe that a metagenomic sequencing approach is warranted in the future to analyze gut microbiomes at genetic and functional levels and compare them at functional and physiological levels. Other factors such as differences in past food experience, genetics, and physiology are also likely to influence eucalypt foraging in koalas (*Blyton et al., 2019*; *Ogura et al., 2019*). Recent studies have reported that when the host and its associated microorganisms are considered as one ecosystem (holobiont), the hologenome, which is the collective term for the host and microbial genomes, can be subject to natural selection (*Zilber-Rosenberg & Rosenberg, 2008*; *Sharon et al., 2010*; *Jia et al., 2021b*; *Wang et al., 2021*). Therefore, future studies should focus on the symbiotic relationship and coevolution between koalas and the gut microbiome to better understand individual and regional differences in eucalypt diet selection by koalas.

There are several limitations in the interpretation of the present results. First, this study used the mitochondrial lineage as the region of origin of koalas. However, the mitochondrial lineage can only divide koalas into four groups, making examining the relationship with regional differences in actual vegetation difficult. A method using the nuclear genome, which allows for finer groupings (*Johnson et al., 2018*) and groupings that consider actual vegetation, would help clarify the role of the gut microbiome in adaptation to regional vegetation. Secondary, as the study subjects were captive koalas, it is possible that koalas may have been affected by the captive environment. It is known that the gut microbiome of herbivorous species is less affected by captive environment (*Delsuc et al., 2014*) and there have been reported that there are few differences in the gut microbiome between captive and wild koalas as well (*Barker et al., 2013*). As in previous studies with wild koalas (*Brice et al., 2019*; *Barker et al., 2013*; *Littleford-Colquhoun et al., 2022*; *Alfano et al., 2015*), the gut microbiome of koalas in this research were dominated by Bacteroidetes (35.97–80.66%) and Firmicutes (8.06–54.88%), followed by Proteobacteria (0.33–26.88%), Verrucomicrobiota (0.00–15.67%), Synergistota (0.07−6.38%) and Cyanobacteria (0.00−6.03%) (Fig. S1). It is also considerable that the difference of the environment such as husbandry facility

and husbandry method (*e.g.*, with or without cohabiting individuals) could affect gut microbiome, however, we could not find a clear effect of these in this research (Fig. S2; although Umi is the member of Awaji Farm Park England Hill, her gut microbiome was more similar to the koalas in Hirakawa Zoological Park. Himawari and Kibou were also housed in cohabitation, but their gut microbiome similarity was not observed). However, the sample size of this study was limited. It is known that the gut microbiome and preferred eucalypts of mothers and their offspring are similar (*Blyton et al., 2022*; *Martin & Handasyde, 1999*). Further validation in larger numbers and unrelated individuals is needed.

Previous research has reported that geographic distance of habitat influences the similarity of the gut microbiome of wild koalas (*Littleford-Colquhoun et al., 2022*). Moreover, this study revealed that captive koalas in Japan have similar gut microbiomes by region of origin. Koalas have been continuously bred and raised in Japan since the early 1980s. In Japan, breeding and cohabitation between mitochondrial lineages have also been conducted. Thus, the fact that regional characteristics in the gut microbiome have been observed even though many generations of koalas have been bred in Japan, far from their habitat, shows how robust the koala gut microbiome of koalas is and indicates that the gut microbiome of koalas has regional variation based on matrilineal inheritance. Given the importance of the gut microbiome for koala foraging and the strong symbiotic relationship between them, we believe that future research should focus on the three-way relationship between koalas, eucalypts, and the gut microbiome to understand koala foraging ecology and to conduct conservation management.

## CONCLUSIONS

This study revealed that the diversity and composition of the gut microbiome of koalas and their eucalypt diet selection differ by regional origin. We also found that some gut bacteria that may influence koalas' eucalypt foraging are associated with both mitochondrial maternal origin and eucalypt foraging patterns, and that the alpha diversity (particularly evenness) of the gut microbiome correlates with foraging diversity in koalas. These differences could result from vertical transmission of the gut microbiome based on maternal transmission and the robustness of the gut microbiome as a hindgut fermenter. These regional differences may also represent an intraspecific variation in koala foraging strategies. Investigating regional differences in eucalypt composition and genetics as well as physiology of koalas is necessary to better understand koala foraging ecology.

## ACKNOWLEDGEMENTS

We thank the staff of Hirakawa Zoological Park and Awaji Farm Park England Hill for their cooperation in collecting samples and sharing individual information, especially Akira Fukumori and Ayaka Ito (Hirakawa) as well as Atsushi Goto, Masato Akai, Izumi Hirayama, and Kakeru Murai (Awaji). We also thank all members of the Ogura Lab (Kitasato University) for their helpful support, especially Chie Kawase, Honami Kikuchi, and Shuntaro Kitamori for their cooperation in fecal sampling, and all members of the

Hayakawa Lab (Hokkaido University) for support and discussion. The authors also thank Hitoshi Suzuki and Shin-ichiro Aiba for their valuable comments on the manuscript.

### Funding

This work was financially supported by JST SPRING #JPMJSP2119 to Kotaro Kondo, MEXT/JSPS KAKENHI (#21KK0106 to Tadatoshi Ogura and Takashi Hayakawa and # 19K16241, #21H04919, and 23H03579 to Takashi Hayakawa), JSPS Bilateral Collaborations to Takashi Hayakawa (Joint Research Projects JPJSBP 120219902), and Hokkaido University Sousei Tokutei Research to Takashi Hayakawa. The funders had no role in study design, data collection and analysis, decision to publish, or preparation of the manuscript.

### Grant Disclosures

The following grant information was disclosed by the authors:
JST SPRING: #JPMJSP2119.
MEXT/JSPS KAKENHI: #21KK0106, #19K16241, #21H04919, 23H03579.
JSPS Bilateral Collaborations: JPJSBP 120219902.
Hokkaido University Sousei Tokutei Research.

### Competing Interests

The authors declare there are no competing interests.

### Author Contributions

- Kotaro Kondo conceived and designed the experiments, performed the experiments, analyzed the data, prepared figures and/or tables, authored or reviewed drafts of the article, and approved the final draft.
- Mirei Suzuki performed the experiments, analyzed the data, authored or reviewed drafts of the article, and approved the final draft.
- Mana Amadaira performed the experiments, analyzed the data, authored or reviewed drafts of the article, and approved the final draft.
- Chiharu Araki performed the experiments, analyzed the data, authored or reviewed drafts of the article, and approved the final draft.
- Rie Watanabe performed the experiments, analyzed the data, authored or reviewed drafts of the article, and approved the final draft.
- Koichi Murakami performed the experiments, authored or reviewed drafts of the article, and approved the final draft.
- Shinsaku Ochiai performed the experiments, authored or reviewed drafts of the article, and approved the final draft.
- Tadatoshi Ogura conceived and designed the experiments, performed the experiments, analyzed the data, authored or reviewed drafts of the article, and approved the final draft.

- Takashi Hayakawa conceived and designed the experiments, performed the experiments, analyzed the data, prepared figures and/or tables, authored or reviewed drafts of the article, and approved the final draft.

## Animal Ethics

The following information was supplied relating to ethical approvals (i.e., approving body and any reference numbers):

Institutional Animal Care and Use Committee of Kitasato University

## DNA Deposition

The following information was supplied regarding the deposition of DNA sequences:

The sequencing data are available in the DDBJ database LC781735–LC781749 (mitochondrial sequence data) and PRJDB16671 (16S short read data).

## Data Availability

The raw 16S microbiome and feeding behavior data are available in the Supplementary Files.

## Supplemental Information

Supplemental information for this article can be found online at http://dx.doi.org/10.7717/peerj.17385#supplemental-information.

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
