# Peer review of "Association of maternal genetics with the gut microbiome and eucalypt diet selection in captive koalas"

_PeerJ, doi:10.7717/peerj.17385_

## Round 0.1 · original submission · Minor Revisions

Both reviewers agree that this is an interesting paper, but have suggestions for improving the manuscript. Please respond to the reviewers' comments and submit a revised version of your manuscript. I would recommend that you specifically pay attention to the comments by Reviewer 1. Please provide a clear justification for using the gut microbiomes of captive animals.

I look forward to seeing your revised manuscript.

·

Basic reporting

This is a well written manuscript that investigates the effects of lineage on koala gut microbiome. This is a current area of research and the application to zoo-housed koalas is novel. The work is well supported with relevant research. However, the citations are not always clearly provided, and this sometimes makes it difficult to identify the source of the information.
There are 8 figures and a table provided in the work. These are professionally formatted and easy to interpret, with only a few small edits required. They provide useful extra information for the project. Raw data and extra, supplemental files are provided that lend relevance to the wider study.

Experimental design

The work is well within the remit of PeerJ and the ethical implications of the work have been taken into account well. There is evidence of an ethical review process.
While the rationale for the study is well defined, there is something of a mismatch between the data that have been collected and the questions being asked. For example, the main question focuses on the effects of genetics (e.g. matrilineal relationships) on gut microbiome. However, many other covariates have not been acknowledged. For example, the location of the population (e.g. which zoo) and housing conditions are likely to affect animal gut microbiomes, and yet they are not considered. This needs to be reflected in the study. There is some use in investigating captive animal gut microbiomes, especially when the microbiomes are available from wild animals for comparison. Please build this into the work in greater detail.
There are also a few methods points to address more clearly, such as diet plans for animals, and selection of individuals for study, alongside methods on assessing preference in terms of eucalyptus.

Validity of the findings

This is clearly a novel and exciting study. There is a good sample size, especially givewn that the study focuses on a species that is rare in captivity. The data are provided alongside the study and the conclusions are provided (however, see the points on experimental design and the relevance of the conclusions that are being drawn.)

Additional comments

This is a very useful study and it could have some meaningful impact in terms of zoo research. However, there are several key areas for the authors to address in order to develop this work further:
1. Zoo study. Please consider the effects of captivity and their potential impact on gut microbiome. There are a plethora of captive - wild studies e.g. in primates that show captivity has an effect. Please evaluate and consider whether your study addresses this point, or lineage effects. Both could potentially be evaluated here.
2. Diets. Please explain the diets for the animals in more detail, alongside the foraging measures. The + scores in the table are unclear. Please could you also explain how individuals for study were selected.
3. Subspecies / ESU. You have mentioned several different ways of categorising koalas into populations, yet it is unclear which is being used in this work. Please provide more background and explanation so that this is clearer in the work. Similarly, evaluate the efficacy of using mtDNA in greater detail.
With these revisions, the work should be in a stronger position for further consideration.

Reviewer 2 ·

Basic reporting

The Authors present a cohort of fecal samples collected from 15 captive koalas from two zoos in Japan. As koalas feed on eucalyptus, which is toxic, and they show strong individual preferences on specific eucalyptus species, the relationship between diet and gut microbiome is of interest for conservation purposes. Here, the Authors investigated the relationship between regional differences, maternal genetics, eucalyptus diet selection and gut microbiome composition. This study has no major concerns and is valuable addition to the field but it needs refinement.

Minor concerns:
• The structure and the referencing of the Figures in the text is at times incorrect. The Authors present 3 paragraphs in their results, but they show 8 main figures. The authors should incorporate the figures discussed in each paragraph as multi-panel figures. In addition, some figures are either not correctly referenced in the text (Figure 3, with no reference to single panels), or they are entirely not discussed in the text (Figure 2C, 2D)
• Replace “maternal origin” with “mitochondrial maternal origin”, to avoid confusion with the later on mentioned maternal microbial origin.
• The Authors should also clearly state that the vertical transmission of the gut microbiome could not be investigated in this study (due to their use of amplicon sequencing), and should clarify that this is a speculation, rather than a result of this study.
• The Authors should consider adding a figure (e.g. heatmap) showing the microbial composition of the koala gut microbiome, annotated with the relevant metadata reported in table 1. This would represent a very useful information for the field, as there are few studies on the koala gut microbiome.
• Clearly state both in the abstract and the main text the number of fecal samples collected in this study and from how many koalas they were sampled from.
• Line 44: “could” instead of “would”
• Line 344: “to be correlated with the diversity” instead of “with the be correlated to diversity”
• Add caption to table 1, describing the metadata columns (e.g. the meaning of foraging + vs -)
• Add caption to Figure 8, specifying what is the meaning of each column of the heatmap
• The Authors should include in the limitations sections that their results on the differential relative abundance of certain taxa might be impacted by the compositionality of the data (see 1)

References:

1 Gloor, G.B., Macklaim, J.M., Pawlowsky-Glahn, V., and Egozcue, J.J. (2017). Microbiome Datasets Are Compositional: And This Is Not Optional. Front. Microbiol. 8)

Experimental design

see comments above

Validity of the findings

see comments above

Additional comments

The authors need to make the raw data available to the public PRIOR acceptance

---

## Round 0.2 · accepted · Accept

Thank you for submitting a revised version of your manuscript. Both reviewers are happy with this version.

·

Basic reporting

The authors have now addressed my main concerns. The manuscript is now stronger as a result and so I would recommend accepting the work for publication.

Experimental design

The authors have now clarified on some of the previous methodological points. The data is clearly illustrated using the graphs.

Validity of the findings

Findings are relevant and the work fills a gap in the knowledge of microbiome science.

Reviewer 2 ·

Basic reporting

The Authors have complied with all requested changes and suggested improvements. The manuscript is ready for publication.

Experimental design

see comment in section 1

Validity of the findings

see comment in section 1